# A Needs Assessment to Inform Research and Outreach Efforts for Sustainable Agricultural Practices and Food Production in the Western United States

**DOI:** 10.3390/foods12081630

**Published:** 2023-04-13

**Authors:** Alexa J. Lamm, Kevan W. Lamm, Sara Trojan, Catherine E. Sanders, Allison R. Byrd

**Affiliations:** 1Department of Agricultural Leadership, Education & Communication, University of Georgia, Athens, GA 30606, USA; kl@uga.edu (K.W.L.);; 2Western Sustainable Agricultural Research and Education, Casper, WY 82601, USA; strojan@sare.org; 3Department of Agricultural and Human Sciences, North Carolina State University, Raleigh, NC 27695, USA; catherine_sanders@ncsu.edu

**Keywords:** Cooperative Extension, sustainable food systems, capacity development

## Abstract

Increasing the adoption of sustainable agricultural practices can help maintain sufficient food production while reducing its environmental impact. To ensure this adoption, it is important to assess the research and training needs of those helping farmers and producers adopt sustainable agricultural practices. However, there is a gap in the literature related to the training needs of producers in the Western United States for sustainable agriculture. Needs assessments help organizations, such as the Western Sustainable Agriculture Research and Education (SARE) program and Cooperative Extension, to address the demonstrated needs of intended audiences. This study presents the results of a needs assessment with the objective of examining training needs and barriers to adoption to help direct extension programming for sustainable agricultural practices in the western region of the United States, to identify gaps, and to inform sustainable agriculture outreach programs. Using a modified Borich method with an inferential statistical method, the discrepancies between the level at which sustainable agricultural practice training competencies “should be addressed” and the level at which they were “currently being addressed” were examined. Competencies with the largest gaps included *financial disparity, food waste,* and *policy/communicating with decision makers*. The top three barriers to adopting sustainable agricultural practices included the potential for *financial loss, perceived risk of adoption*, and *time investment associated with adoption*. Results indicated that training needs varied and that these were not all on-farm training needs. The results imply that future funding from Western SARE and other groups looking to support sustainable agricultural food system efforts, may wish to focus on requesting proposals for programs that address these competency gaps and barriers in novel and supplementary ways in combination with existing programmatic efforts.

## 1. Introduction

The adoption of sustainable agricultural practices to ensure adequate food production while reducing its environmental impact is paramount in a world increasingly focused on feeding a growing population and the water–food–land nexus [1,2,3]. Sustainable agricultural practices have been touted as a solution for minimizing the negative impacts of some conventional agricultural practices [4] as well as a component of achieving the United Nations’ (U.N.) Sustainable Development Goals (SDGs) [5]. In fact, recent work has highlighted the central role of the food system in achieving the SDGs globally [6]. Minimizing negative impacts from agriculture is essential for sustainable development as the food system not only influences the livelihoods of individuals, but also impacts the global environment, politics, and economies [1,3]. The agriculture industry has been called to intensify sustainable production to feed a growing population while minimizing its environmental impact [2] to meet SDG 2, which focuses on ending hunger, achieving food security, improving nutrition, and promoting sustainable agriculture [7]. Sustainable agriculture has been defined as preserving natural resources by balancing economic, social, and environmental costs while supporting communities through maintaining profits for agricultural producers [4,8]. The importance of supporting biodiversity in agriculture is key for maintaining a resilient food system under the effects of climate change [5]. Thus, exploring the research and training needs of the organizations helping farmers and producers adopt sustainable agricultural practices is important for bolstering a resilient food system, especially as modern food production systems face societal and environmental pressures to increase sustainability [9].

The Sustainable Agriculture Research and Education (SARE) program, supported by the United States Department of Agriculture’s (USDA) National Institute of Food and Agriculture (NIFA), aims to address the need for sustainable agricultural development. Driven by producer insights and actions, this “decentralized competitive grants and education program” operates in each United States (U.S.) state and territory [10]. With the goal of aiding all producers in growing safe and abundant food, the organization engages in rigorous reviews of projects, involves farmers in research, and utilizes a multi-disciplinary approach to ensure each funded project includes components for education and outreach. SARE is divided into four regions: north central, northeastern, southern, and western. The current study examines the needs for research and outreach for sustainable agricultural and food production practices in the western region.

An entity that promotes sustainable agricultural development in many states is the Cooperative Extension Service [11,12]. Cooperative Extension Services (hereby referred to as Extension) are part of a national system under USDA NIFA which partners with the land-grant universities in each state to disseminate evidence-based information generated through university research efforts to the public [10,13]. Extension can provide unique forms of assistance for alternative food networks and sustainable agricultural development [14,15], including, but not limited to, providing technical assistance through increased knowledge of agriculture production systems, orchestrating collaboration for new types of community markets, facilitating management of organizational activities, sharing information on securing land access, and educating about mechanisms for profitability and business activities [13]. These issues and needs can serve as potential areas for program development related to sustainable food system development. However, before program development, the needs of stakeholders and clientele should be identified in order to more effectively allocate resources and prioritize efforts [16,17,18].

### 1.1. Study Framework

A needs assessment is “a systematic process of analyzing gaps between what [clients] should know and do” [19], that allows researchers and Extension personnel to identify needs within a specific priority audience and develop programs to meet identified needs [16,17]. Needs assessments are an important aspect of program planning and can provide educators with the ability to learn more about the present conditions and specific needs of an audience by focusing on gaps or deficiencies [19]. Within a needs assessment, researchers and practitioners can measure the discrepancy, or gap, between two conditions—what should be within an organization and what is, or the status of the organization [18]. Extension personnel and other organizational professionals conduct needs assessments by involving multiple groups concerned with the proposed educational program, including learners, educators, community members, and other stakeholders depending on the context of the situation evaluated [19,20]. The current study utilized a needs assessment for the Western SARE organization, whose organizational leadership was interested in better understanding information gaps and future directions for research and outreach efforts related to sustainable agriculture.

Western SARE works with a variety of stakeholders in the Western U.S. to increase knowledge and technology necessary for producers to implement sustainable agricultural practices, including Extension professionals. Extension professionals are uniquely positioned to leverage their credibility in existing agricultural social networks [21] to address the varied and precise needs and concerns of producers [22], and must tailor programming to address the sustainability needs of their targeted client groups based upon the specific barriers to adoption their clients face [23]. The adoption of innovations (including sustainable agricultural practices) is impacted based on various cultural, societal, economic, political, and behavioral aspects [24]. Producers are faced with economic, educational, and social obstacles that prevent their adoption of sustainable practices [25], and these issues also vary according to producer attitudes and experiences [26]. The sustainability knowledge needed for adoption is variable across regional location, type of crop or livestock with which producers work, and existing attitudes [21,26], while regional differences may also be attributed to cultural differences or access to information [27]. Thus, both training needs and barriers to adoption were explored in the current study to help direct Extension programming for sustainable agricultural practices in the western region of the U.S.

### 1.2. Study Purpose

According to Hofman-Bergholm [5], “Sustainability is an intricate concept, and it is strongly connected to fundamental values” (p. 9). Therefore the role of geography and associated values amongst individuals within specific geographies is an important area of inquiry. For example, localizing efforts around the United Nations Sustainable Development Goals [7], has been proposed as a way to ensure sustainable development is relevant to the local needs and aspirations. The research problem is a lack of empirical analysis focused on specific sustainable agriculture topics and barriers to sustainable practice adoption. The purpose of this study was to analyze sustainable topic needs and barriers within the Western United States. The study objectives were very praxis and utility focused. Objective one was to describe the perceived importance of sustainable agricultural practices as well as the extent perceived needs are currently being addressed within Extension. Objective two was to describe the barriers to implementing sustainable agricultural practices.

## 2. Materials and Methods

A quantitative survey was used to collect data through a web-based survey instrument via Qualtrics. The current study was part of a larger needs assessment of gaps and barriers for sustainable agriculture in the western region for both Extension and non-Extension Western SARE stakeholders; however, the current study focused solely on Extension personnel in the western region. Consequently, the methods and data collection descriptions in the current study were applicable to the broader research project. The current study focused on a specific population, Extension personnel.

### 2.1. Instrument Development

The survey was developed in collaboration with Western SARE leadership, Western SARE state representatives, and the Western SARE Administrative Council. The competency items included in the instrument were developed based on a panel of experts composed from the aforementioned groups. The instrument was developed as part of a collaborative needs assessment to determine the current and future research and training needs related to sustainable agriculture for non-profit and non-governmental agencies in the Western U.S. The current study focused only on respondents who worked in Extension.

The instrument was designed using a modified Borich approach [28,29] as well as descriptive statistics. The Borich method is frequently used in agricultural education and Extension research to identify training needs for Extension professionals [30,31]. The Borich method allows for participant self-assessment of various competencies, such as proficiency for sustainable agricultural practices, and their perceived level of importance of the competencies alongside their perceived level of attainment [28]. Additionally, a series of 17 barriers clientele face in adopting sustainable agricultural practices were presented to respondents. Respondents were asked to rate the 30 sustainable agricultural practices on a five-point, Likert-type scale (1 = Strongly Disagree; 2 = Disagree; 3 = Neither Agree nor Disagree; 4 = Agree; 5 = Strongly Agree), both the degree to which the practice *should be addressed* (perceived importance) and are *currently being addressed through research and/or education* (perceived attainment). Respondents were also provided an I don’t know option. Respondents were asked to indicate their level of agreement with a list of 17 barriers associated with clientele adopting sustainable agricultural practices using a five-point Likert-type scale (1 = Strongly Disagree; 2 = Disagree; 3 = Neither Agree nor Disagree; 4 = Agree; 5 = Strongly Agree). Respondents were provided an I don’t know option.

The 30 sustainable agricultural practices and 17 potential barriers were identified by an expert panel based on both domain and application expertise [32]. The expert panel feedback was used to establish content validation for the instrument based on recommendations within the literature [33]. Alternative sources of sustainable practices and barriers were considered based on a review of the literature (such as Gomiero et al. [34]); however, the project was focused on praxis and thus practitioner expertise was deemed to be more applicable to the study intent. To establish response process validity [33], the survey was pilot tested with a group of Extension professionals in the state of Georgia to ensure reliability and face validity. There were minor wording updates to improve readability; however, the overall structure and content of the instrument was deemed valid based on the obtained pilot feedback. The final survey design was deemed exempt by the University of Georgia Institutional Review Board.

### 2.2. Data Collection

The sample for the study included Extension personnel from Western SARE states, including: Alaska, Arizona, California, Colorado, Hawaii, Idaho, Montana, Nevada, New Mexico, Oregon, Utah, Washington, and Wyoming. The sample frame was established by reviewing the email address information available online in the respective states. Additionally, requests were sent to Extension administrators in the represented states and territories requesting email addresses of appropriate respondents. The survey was distributed to 3781 potential respondents within University Extension/Cooperative Extension Services in the Western SARE states.

Data collection occurred using the tailored design method [35] and included a pre-notice message, a response invitation, and three reminder messages sent approximately once per week. All email response requests were sent directly by the research team to potential respondents except Extension professionals at Montana State University (MSU). Potential respondents were provided with a personalized link to the survey so their response could be tracked through Qualtrics and they only received follow-up notices if they had not yet completed the survey. MSU distributed identical messages to those sent by the research team through an internal listserv within the university. Data collection occurred from 14 September to 6 October 2021. A total of 992 respondents who worked for Extension completed the survey for a response rate of 26.2%.

### 2.3. Data Analysis

Data were analyzed via SPSS 27. Surveys with at least 50% completion were included in the analysis. Descriptive statistics were run including frequencies, percentages, means and standard deviations, and inferential statistics included a *t*-test. A group score was calculated for each sustainable agriculture practice by taking the *should be addressed* group mean minus the *are currently being addressed through research and/or education* group mean, modifying Borich’s [28] method of using a mean weighted discrepancy score for each response item [36,37]. Generally, the Borich method helps contextualize the perceptions of a group of respondents to reduce bias or error in one respondent’s perception about an issue through the mean weighted discrepancy score [36]. However, due to the high number of respondents in the survey, the authors used the group mean scores for the *should be addressed (*should*)* and *currently being addressed through research and/or education (*currently*)* groups to conduct a paired sample *t*-test to determine if there were significant differences between group means [38]. The modified approach was employed to more effectively generalize the results to a broader population than is typically accessible through the traditional Borich method [39]. Cohen’s *d* was calculated to assess the effect size. Effect sizes were interpreted based on proposed thresholds within the literature, including small (*d* = 0.2), medium (*d* = 0.5), and large (*d* = 0.8) [40].

Within the barriers to adoption analysis, an adaptation to the feature significance index (FSI) proposed by Gaworski et al. [41] was employed. Specifically, the analysis was adapted to serve as a barrier significance index (BSI). Based on recommendations in the literature [41], the BSI was calculated by dividing the percentage share of agree and strongly agree ratings (4 and 5) by the percentage share of strongly disagree and disagree (1 and 2) ratings using the following formula:(1)$$BSI=\frac{ps_{4,5}}{ps_{1,2}}$$

The BSI provides a ratio of positive to negative responses associated with the barrier. A higher BSI value indicates a larger perceived barrier. The BSI values were then plotted to visually represent the distribution of responses.

### 2.4. Respondent Demographics

The demographic characteristics of respondents can be seen in Table 1. The largest portion of respondents (32.4%) were male, 53.3% identified as white, and were between 30 and 69 years of age. Responses were obtained from all states and territories contacted.

## 3. Results

Descriptive statistics for items representing *should be addressed (*should*)* and *are currently being addressed through research and/or education (*currently*)* needs identified as important by Extension professionals in the Western U.S. are presented in Table 2. Items were ordered based on the difference between the mean level at which items were currently being addressed minus the mean level at which items should be addressed. All items had a negative difference, indicating the perceived need to which the item should be addressed was greater than the level at which the item was currently being addressed. Financial disparity had the largest observed difference between the level at which it was currently being addressed and the level at which respondents felt it should be addressed. Agronomy including crop production had the smallest observed difference between the level at which it was currently being addressed and the level at which respondents felt it should be addressed. A paired samples *t*-test was run for each item, comparing the level at which respondents felt the need should be addressed and the level at which it was currently being addressed. All items were observed to be statistically significant (*p* < 0.001) and all items had a large effect size, reported through Cohen’s *d* (see Table 2).

When respondents were asked to indicate their level of agreement or disagreement with 17 barriers clientele face in adopting sustainable agricultural practices, the respondents agreed that the potential for financial loss was the most pressing barrier for clientele adoption followed by the perceived risk of adoption and time investment associated with adoption. The barriers perceived as least impactful overall were the language barriers and the negative attitude toward sustainable agriculture (see Table 3).

A visual representation of the calculated BSI scores is provided in Figure 1.

## 4. Discussion

A growing global population coupled with the increasing effects of climate change on food systems demand that agriculturalists increase production while reducing negative environmental outcomes [1,2]. Given that sustainable agriculture has become a high-priority research and outreach area within both the U.S. and U.N. agricultural research agendas, it is imperative to conduct needs assessments to understand the gaps in priorities and actions currently being addressed by Extension. Extension professionals have a unique role to play in leveraging the adoption of sustainable agricultural practices and building partnerships between stakeholders [13,14,15], as in the U.S. they have credibility among agricultural producers [21] to provide resources on sustainable production tailored to their unique clientele in diverse U.S. regions [22].

Needs assessments can help create effective programming for expanding the adoption of sustainable agricultural practices within the western region of the U.S. [19]. Through understanding the gaps between what Extension personnel think should be addressed compared to what is currently being addressed, resources, such as those provided by Western SARE, can be allocated appropriately within the region with the necessary policies implemented to support strategic development of needed programs.

### 4.1. Sustainable Agriculture Topic Needs and Status

Previously, Gomiero et al. [34] provided several potential actions that may result in more sustainable agricultural practices. Specifically, the authors identified: agroecology, agriculture intensification, integrated agriculture, organic agriculture, permaculture, precision agriculture, perennial crops, and transgenic technology as potential focus areas. The results of the present study provide an empirical analysis of specific sustainable agriculture topics within a specific region of the United States. The five items with the smallest gap between perceived importance (should) and the level at which it was being addressed (current) were pesticide safety, food safety, food preservation, livestock production, and agronomy including crop production. This finding implies that these issues, while perceived as important, are perhaps being addressed at an appropriate level through Western SARE programming and funding. Therefore, there may not be as pressing a need for new programs and resources in these areas. Rather, continued support for the work conducted in these areas currently is recommended. Based on existing capacities in these areas, Western SARE may consider increasing communication about the work currently happening. Increased communication may help facilitate knowledge dissemination within the region and increased adoption of associated sustainable practices [21,26].

The results also indicated the biggest gap between perceived importance (should) and level at which it was being addressed (current) was within financial disparity, followed by food waste, policy/communicating with decision makers, social disparity, and climate smart agriculture items. The results imply future funding from Western SARE and other groups looking to support sustainable agricultural food system efforts, may wish to focus on requesting proposals for programs that address these concerns in novel and supplementary ways in combination with existing programmatic efforts. The results also imply the role and necessity of evaluating the cultural, societal, political, and behavioral influences that might influence the adoption of sustainability efforts [24] within a specific region. When compared by level of importance (should) and level at which it was being addressed (current), all items were identified as statistically significant, indicating the need for programmatic development across a range of sustainable agricultural practice within Extension in the western region of the U.S.

These results provide an empirical supplement to previous sustainability research in the literature [34]. The items associated with the present study were derived from a panel of experts with experiences and observations directly related to sustainable agricultural production practices. What these results provide are a benchmark for future needs assessments related to sustainable agricultural production, as well as the importance of context relevant areas of focus.

### 4.2. Barriers to Sustainable Agricultural Practice Adoption

Overall, the primary adoption barriers identified by Extension professionals inhibiting producers in the Western U.S. were those associated with risk, finances, and time. These observed results imply that Extension professionals should focus on providing producers with information on programs available to mitigate the risk and financial burden associated with implementing sustainable agricultural practices, such as grant programs for which they qualify. Extension professionals may also consider connecting producers with innovators who have successfully implemented projects on agricultural operations similar to their own in order to reduce the perceived risk and time associated with trial-and-error of adopting practices.

When analyzing the results associated with the barrier significance index (BSI), there are several additional observations of note. First, across 15 of the 17 items, there was a positive ratio observed between the percentage of respondents who agreed that the barrier was relevant versus respondents who disagreed, saying the barrier was not relevant. This finding implies the majority of barriers were considered important. Only two items, a negative attitude toward sustainable agriculture and language barriers, were not observed to be important according to the calculated BSI values. The remaining 15 items had observed BSI scores ranging from 17.72 for perceived risk of adoption to 1.11 for lack of access to current sustainable agriculture information. From a heuristic perspective, the BSI values provide an empirical guide to inform priorities and items that are deemed most important.

### 4.3. Limitations

The results of the current study provide a novel and geographically unique perspective on sustainable agricultural production. However, it is important to also acknowledge the limitations associated with the research. First, the items associated with the study, both sustainable agricultural practices and barriers to sustainable agricultural practice adoption, were developed based on input from a panel of experts. Therefore, the results of the study are limited to the items that were included in the instrument. Consequently, interpretation of the results should be limited to the items that were analyzed.

From a similar perspective, the respondents associated with the study were limited to Extension professionals. Although Extension professionals are generally perceived to be closely associated with, and aware of the needs of, clientele (see [23]), the results should only be interpreted as Extension professional perceptions. A recommendation would be to use the study results as a benchmark for future research and analysis with other audiences, geographies, and so forth.

### 4.4. Contributions to Practice and Recommendations

Although the importance of sustainable agricultural production is well established (e.g., [3,7]), there is limited empirical research that has assessed the needs associated with sustainable practices, nor the barriers to adopting such practices. The current study addresses this gap and provides empirical data to indicate which areas are most in need of additional investigation, and which areas are perceived to be of lower need. Based on these results, a recommendation would be to examine the amount of time and financial investment producers perceive to be involved in adopting sustainable practices compared to the actual time and investment required for adoption. The results of such research may help producers understand the realistic benefits and barriers of switching practices, especially given the availability of federal sustainability grants through Western SARE and other available programming.

Additionally, the study provides a robust benchmark for future applied research involving sustainable agricultural practices. These results are also noteworthy from a policy perspective. Establishing empirical benchmarks provides both status data, as well as future oriented target data. Extension professionals may be uniquely positioned to participate in this type research and provide information about the reality of sustainable agricultural practices given the wide array of services they offer as liaisons of land-grant universities to deliver evidence-based information about such practices [13,14,15]. An additional recommendation for policy would be to conduct longitudinal studies tracking needs and changes over time to ensure that programmatic needs are driving funding and programmatic decisions. The novelty of the methods presented here also provide a template for other organizations working in sustainable agriculture to address gaps in training and practice needed to promote sustainable agricultural development. Exploring gaps between the level of perceived need and the level to which need is addressed allows professionals with baseline data to determine priority areas for development, allowing for efficient distribution of time, resources, and personnel.

Future research is also recommended to replicate similar needs assessments across other SARE regions and in other parts of the world to achieve a collective mindset and targeted interventions that will ensure sustainability in our food systems. Such analysis would allow researchers and practitioners to compare gaps between, as well as within, regions and to determine whether currently observed results are unique to the Western U.S. states and territories or are national/international issues. If other areas have smaller gaps or positive differences between perceived importance (should) and level at which the practice is currently being addressed (current), Extension programs in the Western U.S. may be able to adopt and/or adapt activities from successful programs from other regions or parts of the world lending on one another’s strengths. As an overall recommendation, future researchers must be careful to consider the cultural and information access differences that might be attributable to regional differences [27] and consider this if comparing results with other areas.

Lastly, the use of the BSI analysis is recommended in future needs assessments, particularly when assessing the relative importance of barriers to adoption. Analyzing not only the absolute distributions of responses, but also the relative proportions of responses, provides a novel approach to future needs assessments. As was observed within the current study, the approach provides a ratio-based value that may be useful when interpreting Likert-type data.

### 4.5. Conclusions

Within the existing literature, there are numerous studies that identify sustainable agricultural practices (e.g., Gomiero et al. [34]); however, there are a limited number of empirical studies that operationalize sustainable practices and analyze where specific needs may exist. The present study addresses this gap in the literature and provides a very praxis-based analysis of sustainable agricultural practices. Specifically, the largest gaps between current practice and perceived need were related to financial disparity, food waste, policy/communicating with decision makers, and social disparity. These results indicate that there are needs within sustainable agricultural practice that extend beyond technical capacities associated with production. These results illuminate the need for social and human capital development in tandem with technical applications, such as agronomy, including crop production, livestock production, and food preservation which had the lowest observed needs. As it relates to barriers to the adoption of sustainable agricultural practices, similar trends are observed. For example, the lack of access to technology and lack of access to proper equipment were both relatively less important than social and human barriers, such as perceived risk of adoption and time investment associated with adoption. Balancing both the technical needs and social and human needs is critical to the sustainable agricultural production.

## Figures and Tables

**Figure 1 foods-12-01630-f001:**
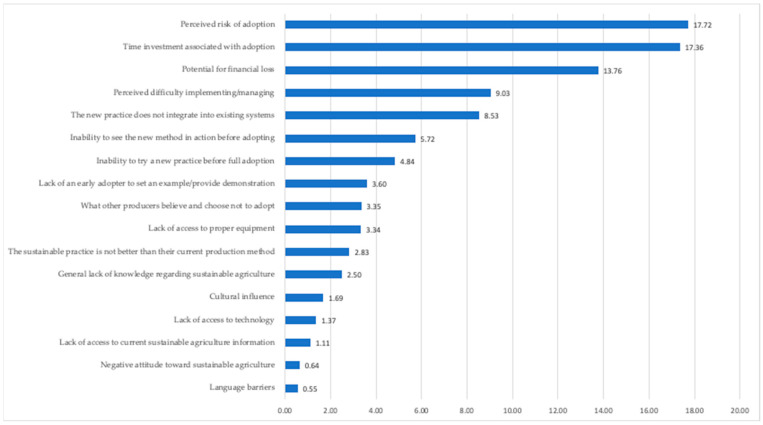
Perceived barriers to adoption BSI analysis.

**Table 1 foods-12-01630-t001:** Demographic characteristics of the respondents (N = 992).

Demographic Characteristic	*F*	%
State		
Washington	200	20.2
California	137	13.8
Oregon	127	12.8
Arizona	111	11.2
Utah	100	10.1
New Mexico	96	9.7
Colorado	82	8.3
Nevada	45	4.5
Wyoming	39	3.9
Idaho	34	3.4
Alaska	14	1.4
Hawaii	4	0.4
American Samoa	2	0.2
Guam	1	0.1
Gender		
Male	321	32.4
Female	291	29.3
Prefer not to answer	41	4.1
No response	339	34.2
Race/Ethnicity		
White	529	53.3
Asian	42	6.5
American Indian	18	1.8
Black/African American	12	1.2
Pacific Islander	4	0.4
Alaska Native	1	0.1
Other	42	4.2
Hispanic/Latinx/Chicanx	56	5.6
Age		
20–29	29	2.9
30–39	118	11.8
40–49	143	14.4
50–59	149	15
60–69	132	13.3
70 and over	34	3.4
No response	387	39.0
Role in Extension		
County Extension Agent/Educator	235	23.7
Extension Specialist	231	23.3
Regional Extension Agent/Educator	59	5.9
Extension Administrator	42	4.2
Other	189	19.1
Years in Extension		
1–10 years	383	38.6
11–20 years	171	17.2
21–30 years	118	11.9
31–40 years	45	4.5
41–50 years	11	1.1
More than 50 years	1	0.1
No response	263	26.5

**Table 2 foods-12-01630-t002:** Research and outreach needs—comparison of “should” be addressed to “currently” being addressed.

Item	Should Be Addressed	Currently Being Addressed	*M_c_* − *M_s_*	*t(df)*	Cohen’s *d*
	*M_s_(SD)*	*M_c_(SD)*			
Financial disparity	4.10 (0.96)	3.01 (1.04)	−1.09	14.86 (396) *	1.49
Food waste	4.10 (0.97)	3.05 (1.04)	−1.05	16.80 (424) *	1.33
Policy/Communicating with decision makers	4.30 (0.79)	3.26 (1.03)	−1.04	17.49 (419) *	1.29
Social disparity	4.07 (0.99)	3.04 (1.05)	−1.03	13.61 (410) *	1.54
Climate smart agriculture	4.40 (0.87)	3.42 (1.11)	−0.98	16.65 (456) *	1.29
Water management	4.68 (0.63)	3.80 (1.04)	−0.88	18.54 (516) *	1.14
Technology access	4.27 (0.81)	3.41 (0.97)	−0.86	15.50 (434) *	1.23
Practice adoption economics	4.05 (0.85)	3.20 (0.91)	−0.85	13.79 (319) *	1.17
Educating about agricultural sustainability	4.39 (0.74)	3.57 (0.98)	−0.82	16.90 (505) *	1.18
Sustainable food systems	4.48 (0.82)	3.70 (1.00)	−0.78	15.24 (505) *	1.18
Marketing/Communicating with consumers	4.12 (0.86)	3.37 (0.95)	−0.75	14.23 (448) *	1.21
Value added processing	4.20 (0.87)	3.48 (1.03)	−0.72	14.50 (421) *	1.17
Agroforestry	3.93 (1.03)	3.21 (1.10)	−0.72	11.57 (393) *	1.26
Federal/State incentives	4.02 (0.87)	3.34 (0.96)	−0.68	11.54 (409) *	1.25
Nutrient management	4.43 (0.73)	3.77 (0.95)	−0.66	14.44 (486) *	1.10
Regulation	4.06 (0.86)	3.43 (0.92)	−0.63	12.34 (446) *	1.16
Natural resource management	4.48 (0.75)	3.87 (0.88)	−0.61	14.30 (489) *	1.04
Soil health and management	4.57 (0.64)	3.96 (0.86)	−0.61	15.47 (550) *	1.00
Agritourism	3.79 (0.93)	3.21 (0.94)	−0.58	10.97 (417) *	1.21
Wildlife management	4.11 (0.86)	3.56 (0.98)	−0.55	11.76 (448) *	1.12
Precision agriculture	4.18 (0.92)	3.65 (1.03)	−0.53	10.41 (456) *	1.20
Aquaculture	3.57 (1.11)	3.05 (1.08)	−0.52	8.30 (369) *	1.22
Range management	4.28 (0.87)	3.90 (0.95)	−0.38	10.78 (465) *	0.97
Organic systems	4.02 (0.94)	3.67 (0.96)	−0.35	7.72 (495) *	1.19
Integrated pest management	4.53 (0.70)	4.19 (0.81)	−0.34	9.69 (544) *	0.93
Pesticide safety	4.39 (0.79)	4.05 (0.89)	−0.34	9.20 (517) *	0.98
Food safety	4.30 (0.80)	3.97 (0.89)	−0.33	8.90 (504) *	0.96
Food preservation	4.02 (0.97)	3.70 (0.96)	−0.32	8.80 (450) *	1.09
Livestock production	4.25 (0.86)	3.97 (0.96)	−0.28	7.03 (477) *	1.12
Agronomy including crop production	4.37 (0.74)	4.11 (0.85)	−0.26	7.38 (518) *	0.98

Note. * *p* < 0.001. *M_s_* refers to the mean scores for should be addressed, *M_c_* refers to the mean scores for currently being addressed.

**Table 3 foods-12-01630-t003:** Sustainable agricultural practices barriers to adoption.

Barrier	*n*	1 (%)	2 (%)	3 (%)	4 (%)	5 (%)	Mean
Potential for financial loss	603	1.00	4.64	16.75	62.69	14.93	3.86
Perceived risk of adoption	601	1.00	3.33	18.97	65.06	11.65	3.83
Time investment associated with adoption	596	0.84	3.52	19.97	65.94	9.73	3.80
Perceived difficulty implementing/managing	603	1.82	6.63	15.26	64.84	11.44	3.77
The new practice does not integrate into existing systems	602	1.33	6.81	22.43	61.13	8.31	3.68
Inability to see the new method in action before adopting	598	1.67	9.87	22.41	57.69	8.36	3.61
Inability to try a new practice before full adoption	590	1.53	11.02	26.78	53.73	6.95	3.54
Lack of an early adopter to set an example/provide demonstration	593	2.02	13.49	28.67	49.41	6.41	3.45
Lack of access to proper equipment	585	2.56	14.19	27.35	49.06	6.84	3.43
What other producers believe and choose not to adopt	591	3.38	12.01	32.99	46.87	4.74	3.38
The sustainable practice is not better than their current production method	589	2.21	15.11	33.62	42.78	6.28	3.36
General lack of knowledge regarding sustainable agriculture	601	3.83	17.64	24.96	48.42	5.16	3.33
Cultural influence	586	6.48	19.97	28.84	40.10	4.61	3.16
Lack of access to technology	596	3.36	24.50	33.89	34.06	4.19	3.11
Lack of access to current sustainable agriculture information	596	5.20	25.67	34.73	31.54	2.85	3.01
Negative attitude toward sustainable agriculture	605	10.08	29.09	35.87	22.48	2.48	2.78
Language barriers	598	10.54	32.61	32.94	21.74	2.17	2.72

## Data Availability

The data are not publicly available due to confidentiality restrictions. Please reach out to the corresponding author for questions related to data availability.

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
