# Peer review of "A Needs Assessment to Inform Research and Outreach Efforts for Sustainable Agricultural Practices and Food Production in the Western United States"

_foods, 2023, doi:10.3390/foods12081630_

Round 1

Reviewer 1 Report

I would like to congratulate the author on the successful completion of an exceptionally interesting study on sustainable agricultural practices and agricultural production. The relevance of the topic is adequately explained in the introduction. The aims and goals of the study are clearly stated. In terms of the methodology, however, I would like to recommend the author consider the following revisions:

1. The author says that the paper is a part of a larger study. Does that mean the author tailored the goals and methodology of that larger study to address the narrower specific tasks? Or was the methodology developed separately? Was the current study an independent element of the larger survey? That should be explained in the Materials and Methods section.

2. How were 3,781 respondents identified? Please detail the approach.

3. Why 30 sustainable agriculture practices and how were they selected? The parameters should be discussed at least briefly. The alternative metrics should be named, their disadvantages in relation to this particular study should be demonstrated.

Conclusion should be separated from the discussion. The results deserve a more comprehensive discussion through the lens of previous studies. In the conclusion (separate section), the author should briefly summarize the findings, the novelty, the implications, the limitations, and potential future research directions.

Author Response

  1. The author says that the paper is a part of a larger study. Does that mean the author tailored the goals and methodology of that larger study to address the narrower specific tasks? Or was the methodology developed separately? Was the current study an independent element of the larger survey? That should be explained in the Materials and Methods section.
    1. ACTION: Thank you for the feedback. We have added additional clarifications to the Materials and Methods section as recommended.
  2. How were 3,781 respondents identified? Please detail the approach.
    1. ACTION: This was a very helpful recommendation. We added additional details regarding the respondent identification process.
  3. Why 30 sustainable agriculture practices and how were they selected? The parameters should be discussed at least briefly. The alternative metrics should be named, their disadvantages in relation to this particular study should be demonstrated.
    1. ACTION: The feedback is understood and appreciated. A description of the practice identification and selection process was added to the methods and materials section along with reference to additional sources which were considered.
  4. Conclusion should be separated from the discussion. The results deserve a more comprehensive discussion through the lens of previous studies. In the conclusion (separate section), the author should briefly summarize the findings, the novelty, the implications, the limitations, and potential future research directions.
    1. ACTION: The reviewer feedback is appreciated. The final section of the manuscript has been rewritten to more appropriately summarize the research, limitations, recommendations, and so forth.

Reviewer 2 Report

About the submission with the title "A Needs Assessment to Inform Research and Outreach Efforts for Sustainable Agricultural Practices and Food Production in the Western United States" I have the following suggestions:

The abstract needs to clarify objetives, methodologias, gaps, novelties and main insights.

I suggest to improve significantly the literature review. 

The methodology should be clearly presented and highlighted the novelties in the approaches considered. In the present version, the methodology seems to much descriptive. Maybe something associated with the factor-cluster analyses, or CART, or ANN, could be more appropriate. 

A section for discussion is missing, where it would be important to benchmark the results obtained with the already published in the literature.

Conclusions section needs to be improved with policy recommendations and practical implications.

In general, for a scientific documents, it seems that something is missing, namely a more robust methodology, or at least highlighting better the novelties of the approaches considered and how they are appropriate for the objectives proposed and for the robustness of the results and conclusions obtained in a scientific research (not a technical report).

Author Response

  1. The abstract needs to clarify objetives, methodologias, gaps, novelties and main insights.
    1. ACTION: We have revised the abstract based on the reviewer’s feedback.
  2. I suggest to improve significantly the literature review. 
    1. ACTION: We have added additional citations from relevant literature from the journal in the hopes this addresses the reviewer’s comment.
  3. The methodology should be clearly presented and highlighted the novelties in the approaches considered. In the present version, the methodology seems to much descriptive. Maybe something associated with the factor-cluster analyses, or CART, or ANN, could be more appropriate. 
    1. ACTION: Thank you for the feedback. Based on a recommendation from Reviewer 3 we have completed an additional barrier significance index value (BSI). The BSI provides a novel method for quantitatively assessing the relative importance of presented barriers.
  4. A section for discussion is missing, where it would be important to benchmark the results obtained with the already published in the literature.
    1. ACTION: The reviewer feedback is appreciated. We have added additional content to the discussion section linking study results to existing literature.
  5. Conclusions section needs to be improved with policy recommendations and practical implications.
    1. ACTION: The reviewer feedback is understood. The final section of the manuscript has been rewritten with additional policy recommendations and practical implications highlighted.
  6. In general, for a scientific documents, it seems that something is missing, namely a more robust methodology, or at least highlighting better the novelties of the approaches considered and how they are appropriate for the objectives proposed and for the robustness of the results and conclusions obtained in a scientific research (not a technical report).
    1. ACTION: The feedback is understood and appreciated. We have added additional narrative in the manuscript to specifically address the scientific novelty of the analysis and the contribution to the scientific literature accordingly.

Reviewer 3 Report

In the Abstract, the Authors wrote what they did as part of the research / study. It would be better to provide such information in the form of a formulated research objective.

Regardless of the formulated general goals of the research/study, presented in four points, it would be worth writing which of these goals were cognitive (scientific) and which were utilitarian (useful) goals. If the scientific and useful goals are clearly defined, then in the Conclusions chapter it is easier to refer to them and determine whether the goals have been achieved and what are the prospects (scope) of further research to be carried out.

Before stating the objectives of the research study, it would be worth formulating the research problem. I think that based on the review of the state of knowledge presented in the Introduction of the article, the authors can easily formulate a research problem. I suggest that in the summary of the state of the art simply write the sentence: "The research problem is ...". The research problem can be linked to the presentation of a gap in the current state of knowledge, which is then translated into the formulation of the research goal / study. It is true that the Authors wrote about information gaps in the final verses on page 2, but this is in relation to the organization (SARE). I think about these gaps, but from a scientific point of view it would be worth writing in a different way so that it was a gap noticed by the authors of the article.

There are no numbered lines in the review version of the article, so this makes it very difficult to refer to specific parts of the article.

In Table 1 at the end, if there is an item "41-50 years" and "50 years or more", then 50 years applies to both groups, and this should not be the case. I propose to write "more than 50 years" in the second case.

I would like to ask whether, before the survey was sent to the respondents, a preliminary study was carried out using the survey to check whether the issues proposed for consideration in the survey are understandable and are not a source of misinterpretation.

The Conclusions chapter in the current version is more of a Discussion of the research results. I suggest changing the title of this chapter. I also propose to develop the Conclusions chapter in a shorter version, where synthetic information summarizing the conducted research will be provided. This version of the Conclusions will be more accessible to the reader.

Research results are presented only in tables. Perhaps it would be possible to present some of the research results in Figures / graphs, which in a much more perfect way affects the imagination of the reader and facilitates the comparison of the results of the analysis. In addition, results such as those presented in Table 3 are difficult to interpret due to the large number of percentage results. At this point, I could propose the use of the "feature significance index (FSI)" to analyze the results of the research presented in Table 3. The idea of this index was presented in the article "Attitudes of a group of young Polish consumers towards selected features of dairy products". The construction of this indicator is based on the relationship of the responses (in percent) with scores 4 and 5 to the responses (in percent) with scores 1 and 2. This creates a single indicator that can be calculated for each issue considered in Table 3. It is easier to compare one indicator summarizing the response scores for each question (issue) rather than such a large number of percentages. The results of the "feature significance index (FSI)" calculations can, for example, be presented in a graph or in some other way.

Citations in the content of the article must be adapted to the Editor's requirements, as in the case of listing publications in References.

I have comments on the publications cited in the article:

- in References there is "Andenoro ... 2016"; this citation is not in the text;

- in References there is "Caravella J. 2006"; this citation is not in the text;

- "Stead, 2019" is quoted twice in the text; this publication is not in References;

- "Bergtold & Molnar, 2010" is quoted twice in the text; this publication is not in References;

- "Rodriquez et al., 2009" is quoted in the text; this publication is not in References. 

Author Response

  1. In the Abstract, the Authors wrote what they did as part of the research / study. It would be better to provide such information in the form of a formulated research objective.
    1. ACTION: We have revised the abstract to highlight the research objective.
  2. Regardless of the formulated general goals of the research/study, presented in four points, it would be worth writing which of these goals were cognitive (scientific) and which were utilitarian (useful) goals. If the scientific and useful goals are clearly defined, then in the Conclusions chapter it is easier to refer to them and determine whether the goals have been achieved and what are the prospects (scope) of further research to be carried out.
    1. ACTION: The feedback is appreciated. Based on this input we have rewritten the study purpose section to be more clear and applicable to the study.
  3. Before stating the objectives of the research study, it would be worth formulating the research problem. I think that based on the review of the state of knowledge presented in the Introduction of the article, the authors can easily formulate a research problem. I suggest that in the summary of the state of the art simply write the sentence: "The research problem is ...". The research problem can be linked to the presentation of a gap in the current state of knowledge, which is then translated into the formulation of the research goal / study. It is true that the Authors wrote about information gaps in the final verses on page 2, but this is in relation to the organization (SARE). I think about these gaps, but from a scientific point of view it would be worth writing in a different way so that it was a gap noticed by the authors of the article.
    1. ACTION: Thank you for the feedback and recommendation. Based on this feedback we have rewritten the study purpose section to be more appropriate and linked to a research problem. Additionally, we have clarified the study objectives and linked the discussion to the objectives.
  4. There are no numbered lines in the review version of the article, so this makes it very difficult to refer to specific parts of the article.
    1. ACTION: Thank you for the recommendation, we added line numbers in the manuscript as suggested.
  5. In Table 1 at the end, if there is an item "41-50 years" and "50 years or more", then 50 years applies to both groups, and this should not be the case. I propose to write "more than 50 years" in the second case.
    1. ACTION: Thank you for the suggestion, we have made this update per feedback.
  6. I would like to ask whether, before the survey was sent to the respondents, a preliminary study was carried out using the survey to check whether the issues proposed for consideration in the survey are understandable and are not a source of misinterpretation.
    1. ACTION: Thank you for the question. Yes, the instrument was pilot tested with a group of Extension professionals from a different region of the United States. We have clarified this process within the Methods and Materials section of the manuscript.
  7. The Conclusions chapter in the current version is more of a Discussion of the research results. I suggest changing the title of this chapter. I also propose to develop the Conclusions chapter in a shorter version, where synthetic information summarizing the conducted research will be provided. This version of the Conclusions will be more accessible to the reader.
    1. ACTION: Thank you for the feedback. The final section of the manuscript has been thoroughly rewritten to be more applicable, integrated, and comprehensive.
  8. Research results are presented only in tables. Perhaps it would be possible to present some of the research results in Figures / graphs, which in a much more perfect way affects the imagination of the reader and facilitates the comparison of the results of the analysis. In addition, results such as those presented in Table 3 are difficult to interpret due to the large number of percentage results. At this point, I could propose the use of the "feature significance index (FSI)" to analyze the results of the research presented in Table 3. The idea of this index was presented in the article "Attitudes of a group of young Polish consumers towards selected features of dairy products". The construction of this indicator is based on the relationship of the responses (in percent) with scores 4 and 5 to the responses (in percent) with scores 1 and 2. This creates a single indicator that can be calculated for each issue considered in Table 3. It is easier to compare one indicator summarizing the response scores for each question (issue) rather than such a large number of percentages. The results of the "feature significance index (FSI)" calculations can, for example, be presented in a graph or in some other way.
    1. ACTION: Thank you for the recommendation and excellent resource. We have recalculated a “barrier significance index” for each barrier in the study and graphed results in a bar chart. The resulting image significantly clarifies perceived barriers.
  9. Citations in the content of the article must be adapted to the Editor's requirements, as in the case of listing publications in References.
    1. ACTION: We appreciate this feedback. We have revised the references in accordance with the journal.
  10. I have comments on the publications cited in the article:
    1. in References there is "Andenoro ... 2016"; this citation is not in the text;
      1. ACTION: We have removed this citation from the manuscript.
    2. in References there is "Caravella J. 2006"; this citation is not in the text;
      1. ACTION: We have added this citation into the text of the manuscript.
    3. "Stead, 2019" is quoted twice in the text; this publication is not in References;
      1. ACTION: We have added citation to references.
    4. "Bergtold & Molnar, 2010" is quoted twice in the text; this publication is not in References;
      1. ACTION: We have added citation to references.
    5. "Rodriquez et al., 2009" is quoted in the text; this publication is not in References.
      1. ACTION: We have added citation to references.

Round 2

Reviewer 1 Report

My Round 1 recommendations are addressed adequately. The manuscript is now improved substantially

Author Response

Thank you for your review!

Reviewer 2 Report

I suggest to highlight the novelty of the methodology and its contribution for the science.

Author Response

Thank you for your revision. We have added verbiage highlighting the novelty of the methodology to the discussion section.